# A Split Luciferase Complementation Assay for the Quantification of β-Arrestin2 Recruitment to Dopamine D_2_-Like Receptors

**DOI:** 10.3390/ijms21176103

**Published:** 2020-08-24

**Authors:** Lisa Forster, Lukas Grätz, Denise Mönnich, Günther Bernhardt, Steffen Pockes

**Affiliations:** Institute of Pharmacy, University of Regensburg, D-93053 Regensburg, Germany; lukas.graetz@ur.de (L.G.); denise.moennich@ur.de (D.M.); guenther.bernhardt@ur.de (G.B.)

**Keywords:** GPCR, dopamine D_2_-like receptors, β-arrestin, GRK, Emerald luciferase, functional assay

## Abstract

Investigations on functional selectivity of GPCR ligands have become increasingly important to identify compounds with a potentially more beneficial side effect profile. In order to discriminate between individual signaling pathways, the determination of β-arrestin2 recruitment, in addition to G-protein activation, is of great value. In this study, we established a sensitive split luciferase-based assay with the ability to quantify β-arrestin2 recruitment to D_2long_ and D_3_ receptors and measure time-resolved β-arrestin2 recruitment to the D_2long_ receptor after agonist stimulation. We were able to characterize several standard (inverse) agonists as well as antagonists at the D_2long_R and D_3_R subtypes, whereas for the D_4.4_R, no β-arrestin2 recruitment was detected, confirming previous reports. Extensive radioligand binding studies and comparisons with the respective wild-type receptors confirm that the attachment of the Emerald luciferase fragment to the receptors does not affect the integrity of the receptor proteins. Studies on the involvement of GRK2/3 and PKC on the β-arrestin recruitment to the D_2long_R and D_3_R, as well as at the D_1_R using different kinase inhibitors, showed that the assay could also contribute to the elucidation of signaling mechanisms. Its broad applicability, which provides concentration-dependent and kinetic information on receptor/β-arrestin2 interactions, renders this homogeneous assay a valuable method for the identification of biased agonists.

## 1. Introduction

The neurotransmitter dopamine exerts its effects via five dopamine receptor (DR) subtypes (D_1_, D_2_, D_3_, D_4_ and D_5_), which are all members of the superfamily of G-protein-coupled receptors (GPCRs) [1,2,3,4,5]. The family of dopamine receptors are classified into D_1_-like receptors (D_1_ and D_5_) and D_2_-like receptors (D_2_, D_3_ and D_4_) according to their preferred G-protein signaling [6]. While D_1_-like receptors predominantly couple to Gα_s/olf_ proteins and stimulate the adenylyl cyclase (AC), thus increasing the intracellular cAMP level [7], D_2_-like receptors associate with Gα_i/o_ proteins, inhibiting the formation of cAMP [8,9,10]. Dopamine receptors are targeted by a variety of pharmacological agents since anomalous dopamine receptor signaling is implicated in numerous neuropsychiatric disorders in the human body such as schizophrenia [11], Parkinson’s disease [12,13], drug addiction [14,15], genetic hypertension [16], bipolar disorder [17,18] and restless legs syndrome [19,20].

Apart from G-protein-mediated signaling, many GPCRs are known to recruit β-arrestin, which is involved in receptor desensitization, internalization processes and also in signaling (β-arrestin-dependent signaling) [21,22,23]. It is generally accepted that phosphorylation of GPCRs by G-protein receptor kinases (GRKs) or protein kinase C (PKC) at specific clusters of serine and threonine residues located in the receptor C-terminus precedes β-arrestin binding [21,24,25,26]. However, β-arrestin recruitment to agonist-activated non-phosphorylated receptors has also been described, but with lower affinity [21]. Furthermore, β-arrestins also participate in receptor sequestration and play a role in desensitization and subsequent resensitization of GPCR responsiveness [22]. The most abundantly expressed arrestins in mammals are β-arrestin1 and β-arrestin2 [27]. Based on their binding preference towards β-arrestins and their behavior during the internalization, GPCRs can be subdivided in two major classes (class A and B) [28]. A precise classification of the respective dopamine receptors according to this model is very difficult due to the complexity of available data [29,30]. However, in terms of D_2_-like receptors, the D_2_R and the D_3_R are frequently described to be phosphorylated by GRKs, resulting in the recruitment of β-arrestins [31,32,33,34], while no recruitment is described for the D_4_R [35,36]. The D_2_ and D_3_ receptors share a high sequence homology [37] but are regulated differently and show different levels of basal phosphorylation [33,34,38]. 

In current drug development of antipsychotics, the need for biased ligands to reduce adverse drug effects is the subject of lively debate. A study by Masri et al. led to the assumption that functionally selective D_2_ receptor antagonists, specifically preventing β-arrestin2 recruitment may lead to new antipsychotics with reduced extrapyramidal side effects, while retaining their therapeutic benefit [39]. Therefore, the functional characterization of potential future drug candidates, with respect to β-arrestin2 recruitment, is of high relevance particularly in the very early stage of in vitro testing. Different assay techniques have been described for investigating β-arrestin2 recruitment in live cells. Commercially available split reporter assays currently used for high throughput screening do not give temporal information about the receptor/β-arrestin interaction, since they require cell lysis [40] or real-time measurements are hampered by relatively long maturation times of the reporter protein (Venus, a variant of yellow fluorescent protein) [41]. A β-arrestin recruitment assay utilizing a transcription factor is the TANGO assay. Here, β-arrestin is fused to a protease, while a transcription factor, which is able to induce transcription of β-lactamase, is C-terminally attached to the receptor via a linker containing the respective protease cleavage sequence. Once β-arrestin gets recruited, the transcription factor is cleaved off, translocated into the nucleus and β-lactamase is expressed. For detection, a substrate is added and the cells need to be lysed [42]. Another approach for the quantification of β-arrestin recruitment to GPCRs is the LinkLight assay using a permuted luciferase reporter [43]. Here the GPCR of interest is fused to a viral protease and β-arrestin is fused to a permuted firefly luciferase containing a protease cleavage sequence. After arrestin recruitment, the permuted luciferase is cleaved and reconstituted to an active enzyme [43]. In transcription-based assays, the obtained signal is prone to amplification and no kinetic information can be gained from this experimental setup. Additionally, there are optimized luciferases available now that show a higher luminescence output and pH independence of the spectra [44]. We aimed to develop a β-arrestin recruitment assay that overcomes the aforementioned limitations, and the split Emerald luciferase complementation technique, first described by Misawa et al. [45], seemed to be appropriate. The employed Emerald luciferase (ELuc) was cleaved into two fragments of which the N-terminal part was fused to β-arrestin2 (referred to as ELucN-β-arr2) and the C-terminal part to the respective receptor (Figure 1).

The ability to perform measurements in living cells allows one to retrieve kinetic information about protein–protein interactions [40,46]. Additionally, the utilized ELuc results in improved sensitivity of the test system, as the signal brightness is increased compared to commercially available test kits [45]. Moreover, this homogeneous assay can be conducted very rapidly without the necessity for any washing or separation step, facilitating the development of high-throughput screening campaigns [45]. To investigate the responses of the D_2_-like receptors upon agonist stimulation, a β-arrestin recruitment assay, based on the split luciferase complementation technique, was established in this study and is described as follows. 

## 2. Results and Discussion

### 2.1. Characterization of the Receptor Fusion Proteins

To verify the membrane expression of the receptor-luciferase fusion constructs and to investigate a potential influence of the receptor modification on ligand affinities, radioligand saturation binding experiments were performed with the radiolabeled antagonist [^3^H]*N*-methylspiperone at the generated D_2_-like receptor constructs. Saturable binding (Appendix A) was found for all of them and the p*K*_d_ values of 10.56 (D_2long_R-ELucC), 10.31 (D_3_R-ELucC) and 9.40 (D_4.4_R-ELucC) at the respective receptor fusion protein were in good agreement with p*K*_d_ values determined at receptors devoid of the luciferase fragment (subsequently referred to as wild-type, cf. Methods) (Table 1 and Appendix A). This confirms that the fusion of the luciferase fragment to the respective receptor did not markedly impair the affinity to the ligand. 

The results from radioligand displacement experiments at the receptor-ELucC constructs, which were compared with the results obtained at the wild-type receptors (Table 2) supported this finding. At all three investigated dopamine receptor subtypes, the p*K*_i_ values of the tested antagonists haloperidol and nemonapride determined at the receptor fusion proteins correspond very well with the affinities determined at wild-type receptors. In the case of the agonist quinpirole and the partial agonist aripiprazole, slight discrepancies but no general pattern was identified. With a p*K*_i_ of 9.25 compared to 8.32, aripiprazole showed a higher affinity to the D_2long_R-ELucC fusion protein than to the wild-type receptor (Table 2). The same observations were made for aripiprazole at the D_3_R with a p*K*_i_ value of 8.9 at the ELucC construct compared to 7.85 at the wild-type receptor (Table 2). However, the data for the D_4.4_R-ELucC and the wild-type D_4.4_R were in very good agreement with each other. For quinpirole, a biphasic displacement curve was observed at the wild-type D_2long_R, yielding a high- and a low-affinity inhibition constant (Table 2), which is in line with published data [47]. By contrast, a monophasic displacement curve was obtained at the ELucC fusion protein with a p*K*_i_ value that was in between the high- and low-affinity inhibition constant determined at the wild-type D_2long_R (Table 2). At the D_3_R and the D_4.4_R, only monophasic displacement curves could be fitted. For both receptors, quinpirole showed a higher affinity to the wild-type receptor. It is known that the use of whole cells has a significant influence, especially on the determination of agonist affinities [48]. Since the binding experiments on the fusion proteins were carried out with whole cells, this could be the reason for the differences rather than the fusion of the luciferase fragment to the C-terminus of the receptor.

Aiming at the development of an assay, that not only allows the measurement of reliable potencies and efficacies but also offers the possibility to conduct live cell measurements as well as kinetic observations of β-arrestin2 recruitment, each transfectant was tested for the feasibility of a real-time experiment. The D_2long_R-ELucC expressing cells showed robust concentration-dependent responses with high signal-to-background (S/B) ratios to stimulation with different agonists when the substrate D-luciferin was added to live cells (Figure 2). As we used the area under the curve (AUC) for the analyses of the data from the β-arrestin2 recruitment assays, no quantitative correlation between the intrinsic activity and the S/B ratio could be made apart from classifying the ligand as a full/partial agonist. Unfortunately, live-cell measurements at HEK293T cells expressing the D_3_R-ELucC showed no β-arrestin2 recruitment. It was previously reported that the D_3_R only recruits β-arrestin2 to a very small extent [33], but by performing lysis-based measurements, reliable results with reasonable S/B ratios (Figure 2) after stimulation with various agonists were obtained. As expected, data correlate with the respective E_max_ values of the tested agonists. In consistency with published data [35], the cells expressing the D_4.4_R did not show any response to agonistic stimulation in either real-time or lytic endpoint measurements. 

### 2.2. Pharmacological Characterization of Dopamine Receptor Ligands in the β-Arrestin2 Split Luciferase Complementation Assay

Standard agonists and antagonists were tested to explore the suitability of the β-arrestin2 split luciferase complementation assay to pharmacologically characterize dopamine receptor ligands of different qualities of action, regarding their potencies (pEC_50_), efficacies (E_max_) or antagonistic activities (p*K*_b_). As agonists, the endogenous ligand dopamine, as well as pramipexole, a widely used drug for the treatment of Parkinson’s disease, and the full agonist quinpirole, were chosen. With *R*-(−)-apomorphine and aripiprazole, a “third generation” antipsychotic drug, exhibiting a unique activity profile, two partial agonists were included in the study as well [49,50]. For defining the efficacy of each compound at the respective receptor, quinpirole was set as the reference agonist (100%), since it shows a higher chemical stability with respect to oxidation compared to the endogenous ligand dopamine.

All agonists showed a time-dependent increase in luminescence in a concentration-dependent manner, which could be converted to concentration-response curves (Figure 3 and Figure 4A). The pEC_50_ values for all agonists determined at the D_2long_R (Table 3) were in very good agreement with data reported in the literature derived from commonly used assays such as [^35^S]GTPγS binding [49] or cAMP assays [51], not differing more than 0.5 orders of magnitude. The endogenous ligand dopamine exhibited full intrinsic activity in the experiment, whereas pramipexole was only able to elicit 86% of the maximal response induced by quinpirole (Table 3). It was previously reported that pramipexole acts as a partial agonist at the dopamine D_2long_R [52]. Aripiprazole appeared as a partial agonist in recruiting β-arrestin2 to the D_2long_R with a very low intrinsic activity (E_max_ = 8 ± 2%) (Table 3). The efficacy of aripiprazole at the D_2long_R is controversial, with publications claiming that it is an antagonist regarding β-arrestin2 recruitment [39] and others describing it as a partial agonist in recruiting β-arrestin2 with efficacies ranging from 47% to 73% depending on the assay [50]. *R*-(−)-apomorphine is reportedly a partial agonist, which we confirmed in our assay with an efficacy of 87% (Table 3). This fits very well with already published data [52]. For the D_3_R, the potencies of all agonistic compounds (Figure 4A) also correlate very well with published data. Dopamine and pramipexole acted as full agonists, whereas *R*-(−)-apomorphine and aripiprazole showed E_max_ values of 91% and 26%, respectively (Table 3). For both compounds, a partial agonism at the D_3_R has been described elsewhere [49,53], with efficacies in a comparable range.

Antagonistic activities (p*K*_b_) of (+)-butaclamol, domperidone, haloperidol and nemonapride at the D_2long_R (Figure 4B) also correlated very well with data described in the literature (Table 3), with the minor exception of *S*-(−)-sulpiride. The same generally applies to the D_3_R (Figure 4B), with nemonapride and (+)-butaclamol showing slight differences (Table 3). A constitutive interaction of the D_3_R with β-arrestin has been repeatedly reported [37,54], which we also observed in our assay, as all antagonists were capable of lowering the arrestin-dependent luminescence signal at the D_3_R below the baseline. Therefore, the set of antagonists was also tested for inverse agonism in the developed assay (agonist mode) as shown in Figure 5. All these ligands exhibited negative efficacy at the D_3_R and potencies, which were comparable with the respective p*K*_b_ values (Table 3).

### 2.3. Influence of GRK2/3 on β-Arrestin2 Recruitment to the D_2long_R and the D_3_R

According to a generally accepted paradigm, G-protein coupled receptor kinases (GRKs) directly link the attenuation of G-protein signaling to arrestin recruitment and therefore play an important role in the desensitization and internalization processes of GPCRs [21]. However, a large body of this knowledge was gained from studies with the β_2_-adrenergic receptor [21] and it has been shown that there are significant differences among GPCRs. In the case of the D_2_-like receptors, it was shown that especially GRK2 and 3 play an important role in these processes [67]. The exact mechanism is not fully understood yet. Therefore we decided to investigate the influence of these kinases in the developed method. Firstly, effects of the selective GRK2/3 inhibitor compound 101 (cpd101) [68] were investigated. The cells co-expressing the ELucN-βarr2 and the D_2long_R-ELucC or D_3_R-ELucC were pre-incubated with the inhibitor at increasing concentrations and concentration-response curves of quinpirole were generated, as displayed in Figure 6. Surprisingly, the inhibition of GRK2/3 in the cells expressing the D_2long_R led to an increase in the luminescence signal to almost 400% (*p* < 0.05) (Figure 6A) and the potency was decreased by almost one log unit (*p* < 0.05). Regarding the D_3_R, the use of cpd101 had no significant effect on the efficacy (*p* = 0.21) or potency (*p* = 0.19) of quinpirole (Figure 6B). Since, for the D_1_R, phosphorylation by GRK2 precedes the association of the D_1_R with β-arrestin2 [69], we constructed an analogous β-arrestin2 recruitment assay for the D_1_R (cf. Methods) as a control. The D_1_R-ELucC construct was validated by radioligand saturation binding and β-arrestin2 recruitment experiments and the results are presented in the Appendix A. All data were in agreement with data from wild-type receptors. Subsequently, the influence of the inhibition of GRK2/3 by using cpd101 was investigated. As expected, inhibition of the kinases led to a concentration-dependent decrease in maximal response induced by the D_1_R standard agonist SKF81297 (cf. Figure 6C) by about 47% (*p* < 0.05) and the potency was only affected to a minor extent (*p* = 0.87). To further unravel the effects of the GRKs, the impact of exogenous overexpression of GRK2 and/or GRK3 on β-arrestin2 recruitment to the D_2long_R and D_3_R was investigated. The HEK293T cells expressing ELucN-βarr2 and D_2long_R-ELucC or D_3_R-ELucC were transiently transfected with a plasmid encoding GRK2 or GRK3. Their response to stimulation with quinpirole was compared to the response of cells that were mock transfected with the empty vector. As illustrated in Figure 7A, GRK2 overexpression in the D_2long_R-ELucC expressing cells led to a slight increase in luminescence signal, although this was not statistically significant (*p* = 0.17). Interestingly, the overexpression of GRK3 led to a marked decrease in luminescence signal (*p* < 0.05) to about 59% of the maximum signal exhibited by the mock transfected cells. This led to the assumption that the increase of the luminescence signal in the experiments with cpd101 (Figure 6) is mainly caused by the inhibition of GRK3. The β-arrestin2 recruitment to the D_3_R (Figure 7B) was not affected (*p* = 0.21) by exogenous GRK2 overexpression, suggesting that endogenous levels of the GRKs are sufficient to ensure β-arrestin2 recruitment or that GRKs are only marginally involved in this process. Additionally, the potency of quinpirole at either receptor was not altered (*p* > 0.05).

Our results regarding the D_1_R are in line with previous findings, confirming that phosphorylation of the receptor by GRK2 initiates or facilitates the interaction of the D_1_R with β-arrestin2 [69]. In contrast to the D_1_R, the involvement of GRK2 in β-arrestin2 recruitment to the D_2long_R is controversially discussed in the literature. It has been reported that inhibition of the kinase activity of GRK2 leads to reduction of arrestin recruitment [70]. However, it has also been reported that GRK-mediated phosphorylation of the D_2long_R is not necessary for β-arrestin association [71] and that GRK2 is constitutively associated with the receptor, whereby D_2long_R signaling is constitutively suppressed [67]. To the best of our knowledge, the contribution of the GRK3 to phosphorylation or trafficking processes of the D_2long_R has not been subject to extensive studies so far. Our findings suggest that the GRK3 somehow hampers the recruitment of β-arrestin2 to the receptor. The inhibitor cpd101 binds to the active site of GRK2/3 and thus blocks the binding of ATP to the enzyme [68]. Since application of the inhibitor led to a marked increase in luminescence signal (Figure 6A), this led us to the assumption that the kinase activity of the enzyme hampers β-arrestin recruitment to the D_2long_R. With respect to the D_3_R, the findings are consistent with earlier publications, in that D_3_Rs only undergo subtle phosphorylation by GRKs and that they are regulated differently than D_2_Rs [72].

The kinetic profiles of β-arrestin2 recruitment to the D_2long_R and the D_1_R under the influence of cpd101 are shown in Figure 8. In both cases, the time courses of the GPCR/β-arrestin interaction with and without cpd101 were similar; only the efficacy was influenced in opposite directions. It is noteworthy that the kinetic course of β-arrestin2 recruitment to both receptors differs markedly. For the D_1_R, it has been described that β-arrestin2 is recruited rapidly, whereas the complex of receptor and arrestin is relatively unstable and already dissociates at the plasma membrane [28]. These findings are reflected by the course of our kinetic measurement, where we observed a steep increase in luminescence signal followed by a rapid decline after reaching a maximum (Figure 8B). This contrasts with the kinetic behavior at the D_2long_R, where the luminescent signal appears to stabilize (Figure 8A), suggesting that there is a more stable interaction between the D_2long_R and β-arrestin2. 

### 2.4. Influence of PKC on β-Arrestin2 Recruitment to the D_3_R

Different studies on the internalization of D_3_ receptors have confirmed that the GRK/arrestin-dependent pathway plays a subordinate role for these receptors, which is consistent with our results described above (cf. Figure 6 and Figure 7). It has been reported that D_3_Rs are mainly internalized after phosphorylation by PKC [38,73]. PKC is known to play a part in heterologous desensitization of GPCRs [38], so we tested whether it contributes to agonist-induced β-arrestin2 recruitment to the D_3_R. We used Gö6983, an inhibitor of different PKC isoenzymes, to abrogate the PKC-dependent phosphorylation of the D_3_R [73]. The cells were treated with increasing concentrations of the inhibitor before the concentration-response curves of quinpirole were recorded. As shown in Figure 9, inhibition of the PKC led to a significant decrease (*p* < 0.05) of the maximum response elicited by quinpirole. Moreover, the potency of quinpirole was decreased when cells were treated with the inhibitor before the measurement, but not with statistical significance (*p* = 0.19). Altogether, these results suggest that PKC-dependent phosphorylation facilitates β-arrestin2 recruitment to the D_3_R.

## 3. Materials and Methods 

### 3.1. Materials

Dulbecco’s modified Eagle’s medium (DMEM) was from Sigma (Taufkirchen, Germany). Leibovitz’ L-15 medium (L-15) was from Fisher Scientific (Nidderau, Germany). Fetal calf serum (FCS), trypsin/EDTA and geneticin (G418) were from Merck Biochrom (Darmstadt, Germany). Zeocin was purchased from Invivogen Europe (Toulouse, France). The cDNAs of the *h*D_2long_R and *h*D_3_R were kindly provided by Dr. Harald Hübner (Department of Chemistry and Pharmacy, Friedrich-Alexander-University, Erlangen). cDNAs of the D_1_R and the D_4.4_R were purchased from the cDNA Resource Center (Rolla, MO, USA). pcDNA3.1/myc-HIS (B) containing the sequence of the β-arrestin2 fusion construct with the N-terminal fragment of the click beetle luciferase was kindly provided by Prof. Dr. Takeaki Ozawa (Department of Chemistry, School of Science, University of Tokyo). The pIRESneo3 vector was a gift from Prof. G. Meister (Institute of Biochemistry, Genetics, and Microbiology, University of Regensburg, Germany). pcDNA-GRK3 was a gift from Robert Lefkowitz [74] (Addgene plasmid # 32,689; http://n2t.net/addgene:32689; RRID:Addgene_32689). If possible, ligands were dissolved in H_2_O (millipore); otherwise in DMSO (Merck, Darmstadt, Germany). (+)-butaclamol, dopamine, Gö6983, pramipexole, quinpirole and SKF81297 were from Sigma (Taufkirchen, Germany), aripiprazole and haloperidol were from TCI Deutschland GmbH (Eschborn, Germany), *R*-(−)-apomorphine, nemonapride, *S*-(−)-sulpiride, domperidone and Takeda compound 101 (cpd101) were from Tocris Bioscience (Bristol, United Kingdom). Pierce D-luciferin was purchased as a potassium salt from Fisher Scientific GmbH (Schwerte, Germany). 

### 3.2. Cell Culture

HEK293T cells obtained as a kind gift from Prof. Dr. Wulf Schneider (Institute for Medical Microbiology and Hygiene, Regensburg, Germany) were cultured in DMEM supplemented with 10% fetal calf serum at 37 °C in a water-saturated atmosphere containing 5% CO_2_. Cells were routinely tested for mycoplasma contamination using the Venor GeM Mycoplasma Detection Kit (Minerva Biolabs, Germany) and were negative.

### 3.3. Generation of Plasmids for Cells Used in the β-Arrestin2 Recruitment Assay

The fusion construct of the respective dopamine receptor and the C-terminal fragment of the luciferase was generated by using the previously described pcDNA4/V5-HIS (B) vector containing the *h*H_1_R-ELucC construct [75]. The sequence of the *h*H_1_R was replaced by the cDNA of the *h*D_1_R, *h*D_2long_R, *h*D_3_R or the *h*D_4.4_R. The cDNAs were amplified by standard polymerase chain reaction (PCR) using gene specific primers and the Q5 high fidelity DNA polymerase (New England Biolabs, Ipswich, MA, USA). The sequences encoding the receptor-ELucC fusion constructs were cloned into the vector by standard restriction and ligation techniques. The quality of the vectors was controlled by sequencing (Eurofins Genomics GmbH, Ebersberg, Germany). 

### 3.4. Generation of Plasmids for Cells Used for Homogenate Preparation

The *h*D_1_R, *h*D_2long_R, *h*D_3_R and *h*D_4.4_R were cloned into a pIRESneo3 vector via Gibson Assembly. The pIRESneo3-SP-FLAG-*h*H_4_R vector, described elsewhere [76], was linearized using standard PCR techniques. Overlaps, complementary to the vector backbone were attached to the dopamine receptors using PCR. Subsequently, receptors were cloned into pIRESneo3 according to the NEBuilder HiFi DNA Assembly Reaction Protocol, resulting in receptors that are N-terminally fused to the membrane signal peptide (SP) of the murine 5-HT_3A_ receptor and tagged with a codon-optimized FLAG tag, subsequently referred to as wild-type receptors. The quality of the vectors was controlled by sequencing.

### 3.5. Generation of Stable Transfectants

HEK293T cells stably expressing the β-arrestin2 fusion construct were generated as previously described [45]. The cells were seeded into a 6-well plate 24 h prior to transfection. For the transfection with the pcDNA3.1/myc-HIS (B) vector encoding the ELucN-βarr2 fusion construct, Fugene HD transfection reagent (Promega, Mannheim, Germany) was used. Cells were incubated with 2 µg of plasmid DNA at 37 °C for 48 h. Before starting with the antibiotic selection, cells were detached with trypsin/EDTA and transferred to a 75-cm^2^ culture flask. G418 at a final concentration of 1000 µg/mL was added to the culture medium until stable growth was observed (for up to 3 weeks). Subsequently, cells were transfected with 2 µg of the pcDNA4/V5-HIS (B) vector encoding the cDNAs for the dopamine receptor fusion proteins (D_1_R-ELucC, D_2long_R-ELucC, D_3_R-ELucC, D_4.4_R-ELucC) as described above with the exception that X-tremeGENE HP (Roche, Basel, Switzerland) was used as transfection reagent. Selection was performed with 400 µg/mL zeocin. Subsequently, a clonal selection was performed with every cell line for high expression of the modified receptor and β-arrestin2 fusion construct. Therefore, stably transfected cells (see above) were seeded on a 15 cm dish at a density of 1000–2000 cells/dish. After 2 weeks, single clones were picked and screened for the highest S/B ratios as described in Figure 2 by using 1µM quinpirole. HEK293T cells stably expressing the wild-type receptors were generated in an analogous manner. Briefly, 2 µg of the pIRESneo3 SP-FLAG- D_1_R/D_2long_R/D_3_R/D_4.4_R vector were used and selection was achieved in the presence of 600 µg/mL of G418.

### 3.6. Preparation of Cell Homogenates

Homogenates were prepared as previously described [77] with minor modifications. HEK293T cells stably expressing the D_1_R, D_2long_R, the D_3_R or the D_4.4_R were grown in 15 cm dishes to 80–90% confluency. Cells were rinsed with ice-cold PBS (137 mM NaCl, 2.7 mM KCl, 10 mM Na_2_HPO_4_, 1.8 mM KH_2_PO_4_, pH 7.4) and detached from the dishes using a cell scraper in the presence of harvest buffer (10 mM Tris·HCl, 0.5 mM EDTA, 5.5 mM KCl, 140 mM NaCl; pH 7.4) supplemented with protease inhibitors (SigmaFAST, Cocktail Tablets, EDTA-free, Sigma-Aldrich, Deisenhofen, Germany). After centrifugation (500 × *g*, 5 min), the D_2long_R expressing cells were resuspended in homogenate buffer (50 mM Tris·HCl, 5 mM EDTA, 1.5 mM CaCl_2_, 5 mM MgCl_2_, 5 mM KCl, 120 mM NaCl; pH 7.4), whereas the D_3_R or D_4.4_R expressing cells were resuspended in Tris-MgSO_4_ buffer (10 mM Tris·HCl, 5 mM MgSO_4_; pH 7.4) and stored at −80 °C. After thawing, the cells were resuspended in homogenate buffer or Tris-MgSO_4_ buffer, and homogenized using an Ultraturrax (on ice, 5 times for 5 s). The homogenate was centrifuged (6 °C, 50,000 × *g*, 15 min), the pellet was resuspended in binding buffer (50 mM Tris·HCl, 1 mM EDTA, 5 mM MgCl_2_, 100 µg/mL bacitracin; pH 7.4) and homogenized using a syringe and needle (i.d. = 0.4 mm). The homogenate was stored in small aliquots at −80 °C.

### 3.7. Radioligand Binding Experiments with Whole Cells

For radioligand saturation binding with whole cells, expressing the developed D_1_R-, D_2long_R-, D_3_R- or D_4.4_R-ELucC fusion constructs, cells were cultured in a 75 cm^2^ flask to a confluency of approx. 80%, detached with a cell scraper and resuspended in L-15 containing 5% FCS. After centrifugation (600 × *g*, 5 min), the cells were resuspended in L-15 medium containing 100 µg/mL bacitracin at a density of 0.15 × 10^6^ cells/mL. The assay was carried out in a final volume of 200 µL/well in 96-well polypropylene plates. The radioligand [^3^H]SCH23390 (D_1_R; specific activity: 81 Ci/mmol, Novandi Chemistry AB, Södertälje, Sweden) was for the D_1_R in a concentration range from 0.04 nM to 4 nM. For the D_2,3,4.4_R, [^3^H]*N*-methylspiperone (D_2,3,4.4_R; specific activity: 77 Ci/mmol, Novandi Chemistry AB, Södertälje, Sweden) was used in a concentration range from 0.025 nM to 1.5 nM for the D_2long_R and the D_3_R or 3.0 nM for the D_4.4_R. After incubation for 60 min (D_2long,3,4.4_R) or 120 min (D_1_R) at room temperature, bound radioligand was separated from free radioligand by filtration through PEI-coated GF/C filters using a 96-well Brandel harvester (Brandel Inc., Unterföhring, Germany). Filters were transferred to (flexible) 1450-401 96-well sample plates (PerkinElmer, Rodgau, Germany) and after incubation with scintillation cocktail (Rotiszint eco plus, Carl Roth, Karlsruhe, Germany) for 5 h, radioactivity was measured using a MicroBeta2 plate counter (PerkinElmer, Waltham, MA, USA). Total and nonspecific data were fitted by the model “one site-total and nonspecific binding” using a hyperbolic curve fit for total binding and linear regression for nonspecific binding. Specific binding was fitted to the model “one site-specific binding”. *K*_d_ values were transformed into p*K*_d_ and means and SEMs were calculated from the respective p*K*_d_ values.

Competition binding experiments with whole cells expressing the fusion proteins were carried out analogous to saturation binding experiments with whole cells as described above. [^3^H]*N*-methylspiperone was applied at a final concentration of 0.06 nM for the D_2long_R and the D_3_R or 0.5 nM for the D_4.4_R. Nonspecific binding was determined in the presence of 2 µM (+)-butaclamol (D_2long_R, D_3_R) or nemonapride (D_4.4_R). Competition binding curves were fitted using a four parameter fit (“log(agonist) vs. response-variable slope”) or a two site fit (“two sites-fit logIC_50_”). Significance of biphasic fitting was tested using the “extra sum-of-squares F Test” provided by GraphPad. *p* values < 0.05 were considered to indicate statistical significance. All calculations were conducted using Prism 8 (Graph Pad, La Jolla, CA, USA).

### 3.8. Radioligand Binding Experiments with Homogenates

Radioligand binding experiments with homogenates were performed as described for whole cells (see above) with minor modifications. For saturation binding experiments homogenates containing the respective dopamine receptor were resuspended in binding buffer (50 mM Tris·HCl, 1 mM EDTA, 5 mM MgCl_2_ and 100 µg/mL bacitracin, pH = 7.4) to a final concentration of 0.3 µg (D_1_R), 0.3 µg (D_2long_R), 0.7 µg (D_3_R) or 0.5–1.0 µg (D_4.4_R) protein/well. Incubation time was 60 min for the D_2long_R, D_3_R and D_4.4_R or 120 min for the D_1_R. Unspecific binding was determined in the presence of (+)-butaclamol (2000-fold excess, D_1_R, D_2long_R, D_3_R) or nemonapride (2000-fold, D_4.4_R). [^3^H]SCH23390 (D_1_R; specific activity: 81 Ci/mmol, Novandi Chemistry AB, Södertälje, Sweden) was used in a concentration range from 0.04 nM to 7 nM for the D_1_R. [^3^H]*N*-methylspiperone (D_2,3,4.4_R; specific activity: 77 Ci/mmol, Novandi Chemistry AB, Södertälje, Sweden) was used in a concentration range from 0.025 nM to 1.5 nM for the D_2long_R and the D_3_R or 3.0 nM for the D_4.4_R. 

For competition binding experiments, [^3^H]*N*-methylspiperone was applied at a final concentration of 0.06 nM for the D_2long_R and the D_3_R or 0.1 nM for the D_4.4_R. Incubation time was 60 min.

### 3.9. Quantification of β-Arrestin2 Recruitment in Live Cells

HEK293T ELucN-βarr2 cells stably expressing the dopamine receptor-ELucC fusion protein were detached from a 75-cm^2^ flask by trypsinization and centrifuged (700 × *g*, 5 min). The pellet was resuspended in L-15 medium supplemented with 5% FCS, HEPES (10 mM), and the cell density was adjusted to 1.25 × 10^6^ cells/mL. Then, 80 µL/well of this suspension were seeded into a white microtiter 96-well cellGrade plate (Brand & Co. KG, Wertheim, Germany) and incubated overnight at 37 °C in a humidified atmosphere. The next day, 10 µL of a 10 mM solution of D-luciferin in L-15 medium was added to each well and the plate was transferred to a pre-warmed (37 °C) INFINITE 200 Pro microplate reader (Tecan, Grödig, Austria). A baseline was measured for 20 min by recording the luminescence of the entire plate for 100 ms per well in 11 cycles. Serial dilutions of the respective agonists or antagonists were prepared in L-15 medium containing HEPES (10 mM) (assay buffer) and warmed to 37 °C prior to addition to the cells. Subsequently, luminescence was recorded for 45 repeats resulting in an overall time period of 1 h. Negative control (assay buffer) and positive control (quinpirole (D_2long_R), full agonist) were included for normalization of the data from the D_2long_R. For measurements performed in antagonist mode, 10 µL of assay buffer were removed from each well before cells were pre-incubated with the antagonist dilutions (10 µL) for 20 min. Antagonists were added simultaneously with the substrate just before starting the baseline measurement. Then, quinpirole (D_2long_R) or SKF81297 (D_1_R) was added at a concentration eliciting 80% of the maximal response and the final read was started. To correct for slight differences in cell counts or amount of substrate added to each well, the mean of the baseline values just before addition of agonists was subtracted from all subsequently recorded values. Additionally, to account for a change of luminescence that might occur over the time-course of the measurement in the absence of agonist, the recorded values of the solvent control were subtracted from all data. For generating concentration-response curves, the AUC after 50 min was used. Data were fitted to the model “log(agonist) vs. response-variable slope (four parameters)”. The p*K*_b_-values were calculated from IC_50_ values according to the Cheng–Prusoff equation [78]. All calculations were conducted using Prism 8 (Graph Pad, La Jolla, CA, USA).

### 3.10. Quantification of β-Arrestin2 Recruitment by Endpoint Measurement

HEK293T ELucN-βarr2 cells stably expressing the dopamine receptor-ELucC fusion protein were prepared 24 h before as described in the preceding section. In agonist mode, 10 µL of assay buffer were added to each well before addition of 10 µL of agonist in different concentrations, resulting in an assay volume of 100 µL. In antagonist mode, cells were incubated with 10 µL of antagonist in different concentrations for 20 min, before quinpirole (10 µL) was added at a concentration eliciting 80% of the maximum response. After incubating the cells with the compounds for 90 min at room temperature, 50 µL of assay medium from each well were replaced by 50 µL of Bright-Glo luciferase assay reagent. Plates were vigorously shaken for 2 min and bioluminescence was recorded for 1 ms per well using an INFINITE 200 Pro microplate reader (Tecan, Grödig, Austria). Data were fitted to the model “log(agonist) vs. response-variable slope (four parameters)”. The p*K*_b_-values were calculated from IC_50_ values according to the Cheng–Prusoff equation [78]. All calculations were conducted using Prism 8 (Graph Pad, La Jolla, CA, USA).

### 3.11. Statistical Analysis 

Statistic differences were analyzed using a t-Test or a one-way ANOVA. All reported *p* values are two-sided, and *p* values lower than 0.05 were considered to indicate statistical significance. All calculations were performed using the SPSS 26 software (IBM, Armonk, NY, USA).

### 3.12. Data Availability

The datasets generated during the current study are available upon request.

## 4. Summary and Conclusions

In this study, we developed a split luciferase complementation β-arrestin2 recruitment assay for the D_2long_ and the D_3_ receptor, which, in case of the D_2long_R, is also applicable in live cells. The hypothesis that the D_4_R does not recruit β-arrestin2 was confirmed [35], as no recruitment was measured at the D_4.4_R. Our assay represents a homogeneous test principle with a cell-permeable substrate, which allows temporal (kinetic) measurements. Combined with the proximal readout and the short incubation time, it represents a significant improvement over the commercially available assays described above. For the D_2long_ and D_3_ receptors, we demonstrated that the assay is suitable for the determination of ligand potencies and efficacies. Furthermore, the test system is able to discriminate between full and partial agonists and to identify inverse agonism at the D_3_R, which makes it a versatile tool for the characterization of dopamine receptor ligands. Although β-arrestin2 recruitment at the D_3_R has played a rather minor role in the literature so far [33], this determination can still be an important parameter for the complete characterization and development of future biased ligands in the field of dopamine receptors. The influence of GRK2/3 and PKC at the D_2long_R, D_3_R, and D_1_R was investigated using different kinase inhibitors, which shows that the assay can also contribute to the deciphering of signaling mechanisms. In summary, this split luciferase complementation assay is a powerful tool for the determination of β-arrestin2 recruitment in dopamine D_2_-like receptors. Thus it represents an important methodological extension for the identification of biased agonists, e.g., in multiparametric analyses, and the characterization of D_2_-like receptor ligands.

## Figures and Tables

**Figure 1 ijms-21-06103-f001:**
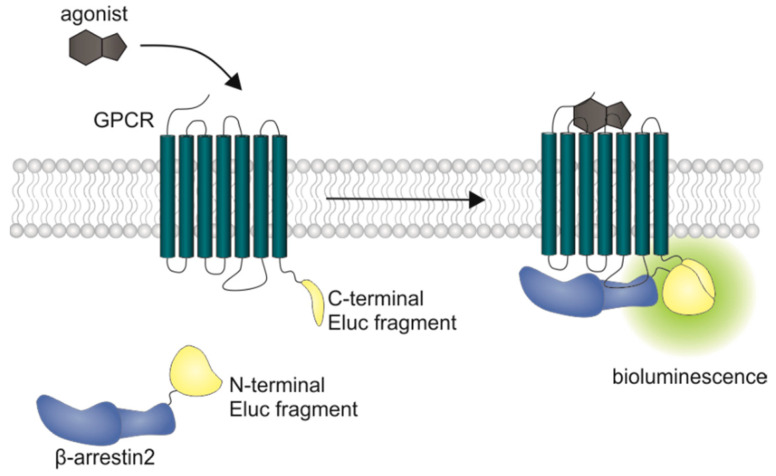
Schematic illustration of the split luciferase β-arrestin2 recruitment assay. Complementary fragments of the Emerald luciferase were fused to β-arrestin2 and the D_1_R, the D_2long_R, the D_3_R or the D_4.4_R. Upon agonist stimulation of the receptor, β-arrestin2 is recruited and the luciferase fragments come into close proximity to form a functional enzyme, which catalyzes the oxidation of D-luciferin to oxyluciferin, accompanied by the emission of light (λ_max_ = 535 nm).

**Figure 2 ijms-21-06103-f002:**
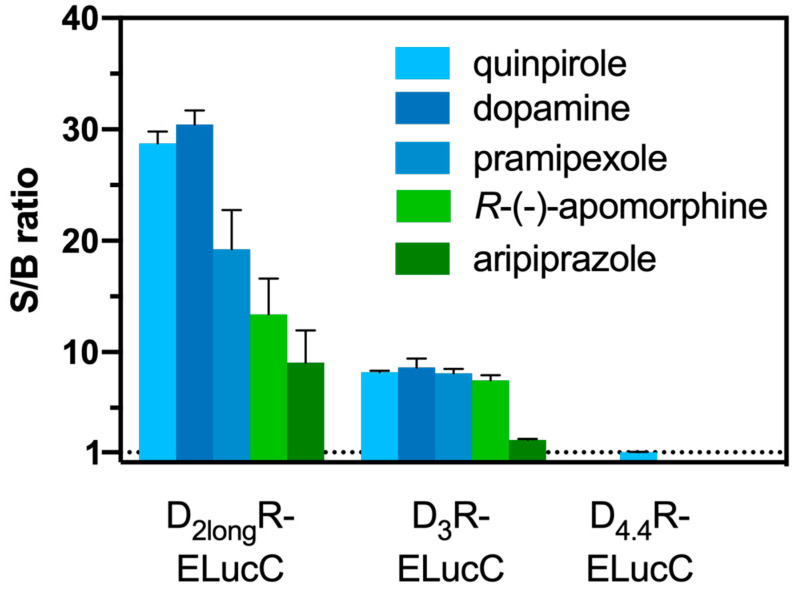
S/B ratios of the complemented ELuc after agonistic stimulation of the analyzed receptors. The cells were stimulated with various agonists (concentration corresponding to the estimated EC_80_ value) and in case of the D_2long_R the resulting AUC after 50 min was divided by the area obtained from a solvent control. S/B ratios of the D_3_R and the D_4.4_R were retrieved by dividing the luminescence after 90 min by that of a solvent control. Data represent means ± SEM from at least three independent experiments performed in triplicate.

**Figure 3 ijms-21-06103-f003:**
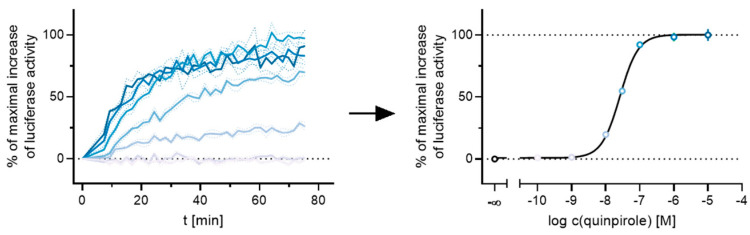
Exemplary results of a live-cell measurement at the D_2long_R. HEK293T cells stably expressing ELucN-βarr2 and D_2long_R-ELucC were stimulated with different concentrations of the standard agonist quinpirole. The time-dependent increase in luminescence was recorded and the AUC after 50 min was used to generate a concentration-response curve.

**Figure 4 ijms-21-06103-f004:**
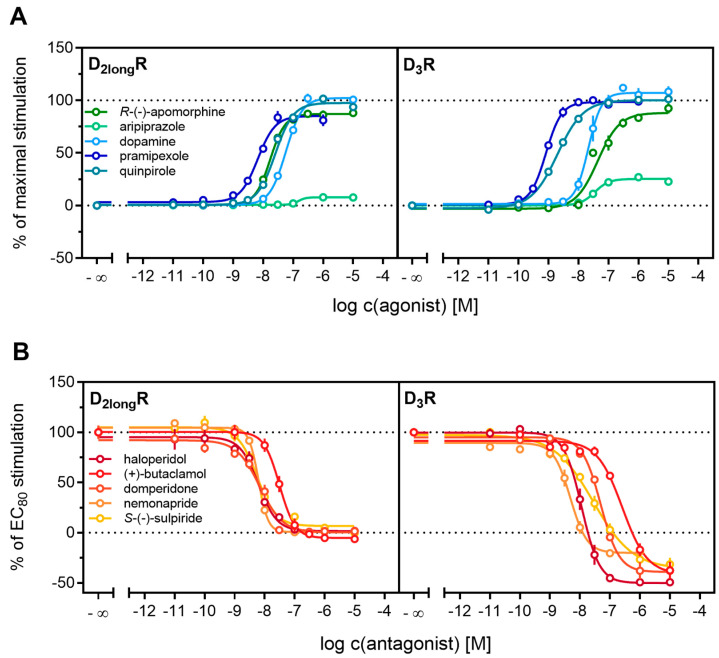
Characterization of standard ligands in the β-arrestin2 recruitment assay. A set of agonists (**A**) and antagonists (**B**) were tested for their ability to promote or inhibit (the quinpirole-induced) β-arrestin recruitment at the D_2long_R and the D_3_R. Data of agonists were normalized to the maximal stimulation induced by 1 µM quinpirole (100%) and a solvent control (0%). Antagonist data were normalized to the signal elicited by quinpirole at a concentration corresponding to the EC_80_ (100%) and a solvent control (0%). Obtained pEC_50_, E_max_ and pK_b_ values are presented in Table 2. Data represent means ± SEM from at least three independent experiments, each performed in triplicate.

**Figure 5 ijms-21-06103-f005:**
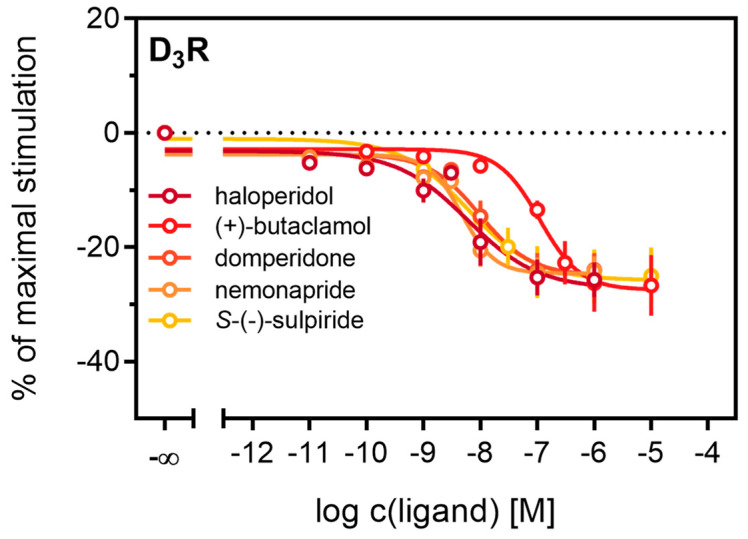
Detection of inverse agonism at the D_3_R. Inhibition of constitutive β-arrestin2 recruitment to the D_3_R by shown D_3_R ligands. Results are presented as percent maximal stimulation as that observed with quinpirole [1 µM]. Data represent means ± SEM from three independent experiments, each performed in triplicate.

**Figure 6 ijms-21-06103-f006:**
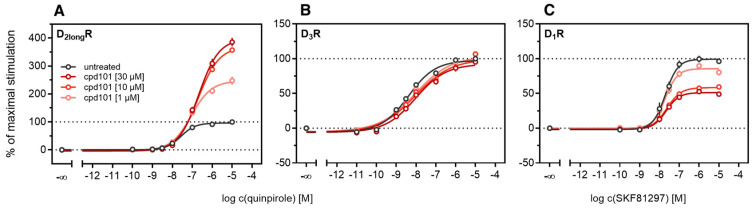
Influence of the GRK2/3 inhibitor cpd101 on β-arrestin2 recruitment. HEK293T cells stably expressing ELucN-βarr2 and the indicated D_x_R-ELucC were incubated with cpd101 at different concentrations for 40 min prior to agonist addition (**A**–**C**). Data represent means ± SEM from three independent experiments, each performed in triplicate.

**Figure 7 ijms-21-06103-f007:**
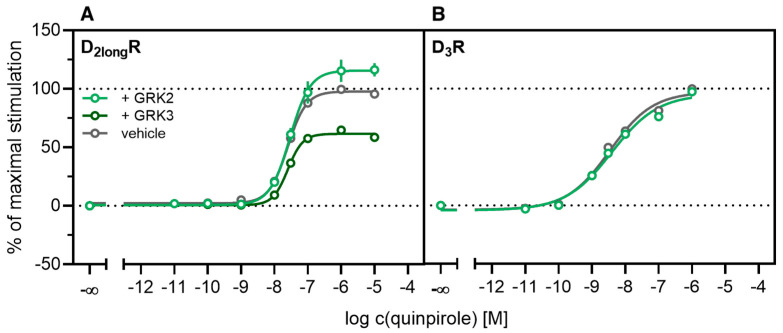
Influence of exogenous GRK2 or GRK3 overexpression on β-arrestin2 recruitment. HEK293T cells stably co-expressing ELucN-βarr2 and the D_x_R-ELucC were transiently transfected with GRK2/GRK3 or empty vector (**A**,**B**). Data represent means ± SEM from three independent experiments, each performed in triplicate or quadruplicate.

**Figure 8 ijms-21-06103-f008:**
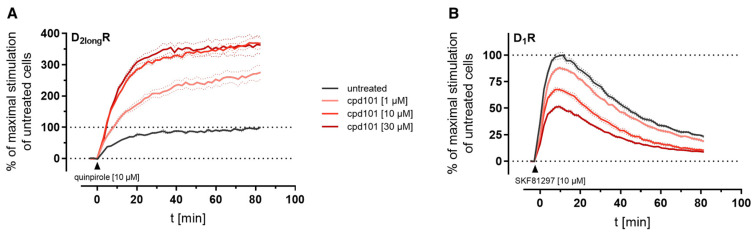
Impact of the specific GRK2/3 inhibitor cpd101 on the kinetics of β-arrestin2 recruitment to the D_2long_R (**A**) and the D_1_R (**B**). Data represent means ± SEM from three independent experiments, each performed in triplicate.

**Figure 9 ijms-21-06103-f009:**
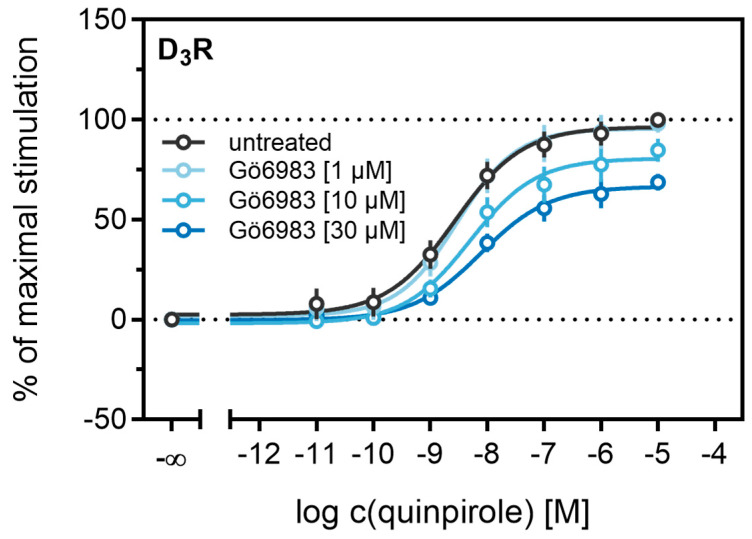
Influence of the PKC inhibitor Gö6983 on β-arrestin2 recruitment. HEK293T cells stably expressing ELucN-βarr2 and D_3_R-ELucC were incubated with Gö6983 at different concentrations 40 min prior to addition of agonist. Data represent means ± SEM from three independent experiments, each performed in triplicate.

**Table 1 ijms-21-06103-t001:** Dissociation constants (p*K*_d_ values) of [^3^H]*N*-methylspiperone determined in radioligand saturation binding experiments at receptors fused to the C-terminal fragment of the Emerald luciferase using whole cells and at wild-type receptors using homogenates. Data represent means ± SEM determined in three independent experiments, each performed in triplicate.

	D_2long_R		D_3_R		D_4.4_R	
	ELucC Fusion Protein	wt	ELucC Fusion Protein	wt	ELucC Fusion Protein	wt
pK_d_	10.56 ± 0.04	10.84 ± 0.05	10.31 ± 0.03	10.59 ± 0.01	9.40 ± 0.09	10.11 ± 0.04

**Table 2 ijms-21-06103-t002:** Inhibition constants (p*K*_i_) of selected standard ligands determined in radioligand displacement experiments. p*K*_i_ values were determined at receptors fused to the C-terminal fragment of the Emerald luciferase using whole cells and at wild-type receptors using homogenates. Data represent mean p*K*_i_ ± SEM determined in three independent experiments, each performed in triplicate.

cpd	D_2long_R			D_3_R		D_4.4_R	
	ELucC Fusion Protein		wt	ELucC Fusion Protein	wt	ELucC Fusion Protein	wt
aripiprazole	9.25 ± 0.16		8.32 ± 0.02	8.9 ± 0.24	7.85 ± 0.08	7.64 ± 0.15	7.85 ± 0.08
quinpirole	7.29 ± 0.07	hi	7.90 ± 0.10	7.63 ± 0.05	8.00 ± 0.08	6.58 ± 0.01	8.00 ± 0.08
		lo	6.11 ± 0.02				
haloperidol	9.45 ± 0.05		9.58 ± 0.13	8.27 ± 0.08	8.93 ± 0.02	8.27 ± 0.02	8.93 ± 0.02
nemonapride	9.95 ± 0.13		9.76 ± 0.08	9.76 ± 0.06	9.33 ± 0.02	9.69 ± 0.08	9.33 ± 0.02

**Table 3 ijms-21-06103-t003:** pEC_50_, E_max_ and p*K*_b_ values of standard compounds analyzed in the newly developed β-arrestin2 recruitment assay. For comparison, p*K*_i_ values determined in radioligand displacement studies utilizing homogenates from HEK293T cells stably expressing the wild-type receptors (cf. Table 2) and published data from different assays are included. Data represent means ± SEM from N independent experiments, each performed in triplicate.

Receptor	cpd	β-Arrestin2 Recruitment			N		Radioligand Displacement	Ref.
		pEC_50_	E_max_ [%]	p*K*_b_			p*K*_i_	
**D_2long_R**	*R*-(−)-apomorphine	7.77 ± 0.04	87 ± 3		4		7.48 ± 0.14	7.66 [49]
	aripiprazole	6.65 ± 0.15	8 ± 2		3		8.32 ± 0.02	6.84 [50]
	dopamine	7.24 ± 0.04	104 ± 3		3	hi	7.99 ± 0.16	7.05 [55]
						lo	6.30 ± 0.07	
	pramipexole	8.19 ± 0.05	86 ± 4		4	hi	7.59 ± 0.12	8.51 [56]
						lo	6.00 ± 0.03	
	quinpirole	7.55 ± 0.07	100		5	hi	7.90 ± 0.10	7.11 [51]
						lo	6.11 ± 0.02	
	(+)-butaclamol			8.29 ± 0.10	3		9.14 ± 0.06	8.04 [57]
	domperidone			9.13 ± 0.09	3		9.47 ± 0.07	8.87 [58]
	haloperidol			8.90 ± 0.05	3		9.58 ± 0.13	8.89 [59]
	nemonapride			8.90 ± 0.05	3		9.76 ± 0.08	9.32 [60]
	*S*-(−)-sulpiride			8.86 ± 0.10	3		7.51 ± 0.09	8.22 [61]

**D_3_R**	*R*-(−)-apomorphine	7.43 ± 0.17	91 ± 5		3		8.40 ± 0.03	7.93 [49]
	aripiprazole	7.44 ± 0.05	26 ± 1		3		8.26 ± 0.02	7.00 [62]
	dopamine	7.66 ± 0.14	105 ± 8		3	hi	8.78 ± 0.09	7.95 [63]
						lo	7.23 ± 0.09	
	pramipexole	9.09 ± 0.06	99 ± 4		4		9.18 ± 0.06	8.65 [49]
	quinpirole	8.75 ± 0.07	100		6		8.34 ± 0.07	9.07 [56]
	(+)-butaclamol	7.16 ± 0.17	−27 ± 9	7.35 ± 0.08	3/3		8.59 ± 0.02	7.95 [64]
	domperidone	8.02 ± 0.14	−26 ± 4	8.06 ± 0.09	3/3		8.96 ± 0.11	8.12 [58]
	haloperidol	8.29 ± 0.29	−27 ± 5	8.68 ± 0.12	3/3		8.95 ± 0.03	8.70 [65]
	nemonapride	8.43 ± 0.13	−25 ± 4	9.07 ± 0.12	3/3		9.99 ± 0.06	9.77 [66]
	*S*-(−)-sulpiride	8.33 ± 0.10	−26 ± 8	8.23 ± 0.07	3/4		7.20 ± 0.03	7.70 [58]

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
