# Peer review of "A Split Luciferase Complementation Assay for the Quantification of β-Arrestin2 Recruitment to Dopamine D2-Like Receptors"

_ijms, 2020, doi:10.3390/ijms21176103_

Round 1
Reviewer 1 Report
This paper presents a split luciferase complementation ß-arrestin2 recruitment assay for the dopamine receptors in live cells. This work is worthy to be published in International Journal of Molecular Sciences after some modifications as suggested below.
(1)
Characterization of D1R (For example, the pKd value of [3H] N-methyl spiperone) should also be included in the manuscript, as the effect of the GRK 2/3 inhibitor cpd101 on the recruitment of β-arrestin2 by D1R is shown in Figures 6 and 8.
(2)
In radioligand saturation binding experiments, the pKd values of ELucC-fused dopamine receptors were in good agreement with those of the respective wild types. However, the absolute amount of the [3H] N-methyl spiperone bound to the three ELucC-fused dopamine receptors is different from that of the wild type (Figure S1). Therefore, please discuss in detail whether luciferase fragment modification to the receptor does not affect ligand binding.
(3)
In addition to quinpirole, the effects of other agonists on S/B ratio of the ELucC-fused dopamine receptors should be investigated and discussed.
Author Response
We thank the referee for his valuable comments.
Comment 1:
As [3H]N-methyl spiperone (suggested by reviewer 1) is solely a suitable radioligand for D2-like receptors, we used the commonly used D1-like receptor radioligand [3H]SCH23390 for our experiments. We made saturation binding experiments (each N = 3) with whole HEK293T ELucN-βarr2 cells expressing the D1R-ELucC and with homogenates from cells expressing the wild-type D1R (Figure S2 & Table S1, Supplementary Material). In addition, β-arrestin recruitment experiments with standard D1R agonists and antagonists (SKF81297 and SCH23390; each N = 3) were performed with whole HEK293T ELucN-βarr2 cells expressing the D1R-ELucC (Figure S3 & Table S2, Supplementary Material). Time for revision was too short to further characterize the receptor in radioligand competition binding experiments, as it was additionally done for the other receptors.
Comment 2:
Modification of the receptor actually did not change binding affinity of the radioligand [3H] N-methyl spiperone markedly, as shown in saturation binding experiments (cf. Table 1). As different receptor preparations were used for saturation binding experiments at the luciferase-tagged receptors or the wild-type receptors, respectively (whole cells vs. homogenates), we did not compare the differences in obtained Bmax values, which correlate with the amount of bound radioligand, directly. The change in receptor concentration did not seem to be relevant for our purposes.
Comment 3:
The effect on S/B ratio of all tested agonists were additionally calculated and presented in Figure 2. The investigations on this were discussed in the relevant section (line 143-168). and adapted in the manuscript. Figure 2 was modified and reinserted in the manuscript.
Reviewer 2 Report
The authors developed the β-arrestin2 split luciferase complementation assay for dopamine receptors (DR) on the base of improved click beetle luciferase. The assay was tested for tree types of DR using selected standard agonist and antagonist ligands and the obtained results were in good agreement with the earlier published results. Moreover, the developed assay allows kinetic measurements of receptor/β-arrestin2 interactions upon ligand binding to DR, including in living cells. The manuscript is technically sound, well written and сovers a topic which is of interest for broad biochemical/biophysical community.
However, the authors should add more details regarding Generation of stable transfectants. It is not clear how the authors obtained the final stable transgenic cell lines and how the expression of the target proteins in them was assessed. I think that antibiotic resistance cannot be a criterion for the expression of the target proteins and a guarantee of the genetic homogeneity of the obtained transfectants.
Also confusing is the writing of the dimension of values along the axes in all graphs of the manuscript after the slash denoting the symbol of division. Maybe it's better to separate the dimension from the name of the value, as usual, with a comma or brackets?
Author Response
We thank the referee for his valuable comments.
Comment 1:
The reviewer is right that antibiotic resistance is no robust criterion for the expression of the target protein. Therefore, a single clone selection for the highest receptor expression was performed as described in the manuscript (added section; lines 404-407). The correct insertion of the receptor in the membrane was assessed by radioligand saturation binding experiments and has already been described in the manuscript. Expression of the b-arrestin fusion construct was verified by the described b-arrestin recruitment assay.
Comment 2:
The reviewer is right. We modified all mentioned Figures (Figure 2-9 & S1) and inserted them into the manuscript. All figures added in the course of the revision were of course formatted analogously.